



# Seasonal evolution of winds, atmospheric tides and Reynolds stress components in the Southern hemisphere mesosphere/lower thermosphere in 2019

Gunter Stober[1], Diego Janches[2], Vivien Matthias[3], Dave Fritts[4,5], John Marino[6], Tracy Moffat-Griffin[10], Kathrin Baumgarten[7], Wonseok Lee[8], Damian Murphy[9], Yong Ha Kim[8], Nicholas Mitchell[11], and Scott Palo[6]

[1]Institute of Applied Physics & Oeschger Center for Climate Change Research, Microwave Physics, University of Bern, Bern, Switzerland
[2]ITM Physics Laboratory, Mail Code 675, NASA Goddard Space Flight Center, Greenbelt, MD 20771, USA
[3]German Aerospace Centre (DLR), Institute for Solar-Terrestrial Physics, Neustrelitz, Germany
[4]GATS, Boulder, CO, USA
[5]Center for Space and Atmospheric Research, Embry-Riddle Aeronautical University, Daytona Beach, FL, USA
[6]University of Colorado Boulder, Colorado, USA
[7]Fraunhofer Institute for Computer Graphics Research IGD, Rostock, Germany
[8]Department of Astronomy, Space Science and Geology, Chungnam National University, Daejeon 34134, South Korea
[9]Australian Antarctic Division, Kingston, Tasmania, Australia
[10]British Antarctic Survey, UK
[11]University of Bath, Bath, UK

**Correspondence:** gunter.stober@iap.unibe.ch

**Abstract.** In this study we explore the seasonal variability of the mean winds, diurnal, semidiurnal tidal amplitude and phases as well as the Reynolds stress components during 2019, utilizing meteor radars at six southern hemisphere locations ranging from mid- to polar latitudes. These include Tierra del Fuego, King Edward Point on South Georgia island, King Sejong Station, Rothera, Davis and McMurdo stations. The year 2019 was exceptional in the southern hemisphere, due to the occurrence of a
rare minor stratospheric warming in September. Our results show a substantial longitudinal and latitudinal seasonal variability of mean winds and tides pointing towards a wobbling and asymmetric polar vortex. Furthermore, the derived momentum fluxes and wind variances, utilizing a recently developed algorithm, reveal a characteristic seasonal pattern at each location included in this study. The longitudinal and latitudinal variability of vertical flux of zonal and meridional momentum is discussed in the context of polar vortex asymmetry, spatial and temporal variability, and the longitude and latitude dependence of the
vertical propagation conditions of gravity waves. The horizontal momentum fluxes exhibit a rather consistent seasonal structure between the stations while the wind variances indicate a clear seasonal behaviour and altitude dependence showing the largest values at higher altitudes during the hemispheric winter and two variance minima during the equinoxes. Also the hemispheric summer mesopause and the zonal wind reversal can be identified in the wind variances.





# 1 Introduction

Gravity waves (GW) are an essential driver of the mesosphere/lower thermosphere (MLT) dynamics forcing a meridional flow due to a zonal drag, which drives the mesopause temperature up to 100 K away from the radiative equilibrium (e.g., Lindzen, 1981; Becker, 2012) introducing a residual circulation from the cold summer to the warm winter pole. This important coupling mechanism is caused by GWs carrying energy and momentum from their source regions to the altitude of their breaking,

coupling different vertical layers in the atmosphere (Fritts and Alexander, 2003; Ern et al., 2011; Geller et al., 2013). The primary forcing of the MLT at small scales is by gravity waves arising from various tropospheric sources, among them flow over orography (mountain waves), deep convection (convective gravity waves), frontal systems, and jet stream imbalances and shear instabilities (Fritts and Nastrom (1992) also see the reviews by Fritts and Alexander (2003) and Plougonven and Zhang (2014). These various GWs typically have horizontal phase speeds comparable to mean winds at higher altitudes, hence are

strongly influenced by varying winds along their plane of propagation. Where they approach a critical level at which their phase speed, ch, along their direction of propagation equals the mean wind, Uh, in this plane, they undergo breaking and dissipation, resulting in local mean flow accelerations that act as sources of secondary GWs (SGWs). GW breaking dynamics lead to SGWs that occur on relatively small horizontal scales, 10-100 km; SGWs at larger scales, 100-300 km, arise due to the local, transient mean-flow accelerations accompanying GW momentum transport (Dong et al., 2020; Fritts et al., 2020). SGWs at

larger scales also arise due to interactions among larger-scale GWs in global models unable to resolve GW breaking dynamics (Becker and Vadas, 2018; Vadas and Fritts, 2001; Vadas and Becker, 2018). Importantly, however, SGWs accompanying GW breaking and interactions at lower altitudes require propagation over large depths to themselves become significant, hence play more significant roles in the lower thermosphere.

Although GWs are such an important driver of the MLT, the number of observations is rather sparse. Very often the GW

activity is inferred by subtracting a background from the wind or temperature observations to estimate potential GW energy or wind variations (Ehard et al., 2015; Baumgarten et al., 2017; Chu et al., 2018; Rüfenacht et al., 2018; Stober et al., 2018b; Wilhelm et al., 2019). Satellite observations provide an estimate of absolute momentum fluxes from the troposphere up to the mesosphere and most importantly a global coverage (Ern et al., 2011; Trinh et al., 2018; Hocke et al., 2019). However, satellite observations are lacking the directional information and, thus, there is some ambiguity about the forcing or whether the GW

momentum flux is accelerating or decelerating the mean flow.

Vincent and Fritts (1987) introduced, over two decades ago, a radar technique to determine the vertical flux of zonal and meridional momentum from Medium Frequency (MF)-radars using two pairs of co-planar beams. This technique was also applied by Placke et al. (2015b, a) to determine momentum fluxes above Andenes in Northern Norway. However, there are only a few MF-radars worldwide that are able to conduct such measurements. Further, at altitudes above 94 km MF-radars tend

to underestimate the wind speeds, which might lead to some systematic bias in the derived momentum flux (Wilhelm et al., 2017).

Meteor radars, on the other hand, are widely distributed around the globe and have been shown to measure reliable winds in the ∼70-100 km altitude range (McCormack et al., 2017). Hocking (2005) proposed a method to obtain the Reynolds stress





tensor components from the meteor radar observations. Based on this, several studies investigated the method to optimize the

data analysis as it appeared to be challenging to get the technique implemented (Placke et al., 2011a, b; Andrioli et al., 2013). Fritts et al. (2010b) presented a momentum flux meteor radar design to overcome some of the difficulties and evaluated the momentum flux observations using synthetic data (Fritts et al., 2010a), which finally provided evidence that the systems can be used to measure momentum fluxes. This led to several studies using these new generation systems (de Wit et al., 2014, 2016, 2017; Spargo et al., 2019; Vierinen et al., 2019) or more powerful radars such as MU-radar (Riggin et al., 2016).

In this study, we show results from a recently developed retrieval algorithm, which builds on the initial momentum flux analysis formulation reported by Hocking (2005). In particular, we introduce a generalized approach to obtain wind variances and momentum fluxes from several meteor radars, many of which are standard low power systems, for the year 2019, which evolved into one of the rare minor stratospheric warming events during September (Yamazaki et al., 2020). We briefly summarize how the Reynolds stress components, also called momentum fluxes and wind variances, are derived from a Reynolds decomposition.

The Reynolds decomposition is achieved by utilizing an adaptive spectral filter (ASF), which allows the decomposition of the meteor radar wind time series into mean winds, tidal components and a GW residual (Stober et al., 2017; Baumgarten and Stober, 2019; Stober et al., 2019) similar to the S-transform used in previous studies (Stockwell et al., 1996; Fritts et al., 2010a). Here we analyze wind observations from six meteor radars operating in the southern hemisphere at mid- and polar latitudes. The meteor radars are located at Tierra del Fuego (Argentina) (TDF) (Fritts et al., 2010b; Liu et al., 2020), King

Edward Point (KEP) on South Georgia island (Jackson et al., 2018), at King Sejong Station (KSS) on King George Island (Lee et al., 2018, 2016), and at Rothera (ROT) Station (Sandford et al., 2010) located on the Antarctic Peninsula, Davis Antarctic Station (DAV) (Holdsworth et al., 2004) and McMurdo (MCM) Antarctic Station. Longitudinal and latitudinal differences of mean winds, diurnal and semidiurnal tides, as well as the momentum fluxes and wind variances are discussed in the context of the wobbling of the polar vortex with increasing altitude up to the MLT and the polar vortex providing a temporally and

spatially variable filter for vertical GW propagation. Additionally, we consider the orographic environment for each station to underline the difference in environment among all the meteor radar sites.

The manuscript is structured as follows: In section 2 we present a brief introduction to the wind retrievals, the derivation of the Reynolds stress components and the implemented momentum flux and GW retrieval. Section 3 contains the results of the mean flow terms, which are mean winds, diurnal and semidiurnal tides and their seasonal behaviour, as well as the determined

momentum fluxes and wind variances in this section. Our results are discussed in section 4 and the conclusions are provided in section 5.

## 2    Observations and methods

### 2.1    Meteor radar observations

In this study, we use observations obtained with 6 meteor radars operating in the Southern Hemisphere between 53°S and 79°S

in latitude. Four of the meteor radars can be grouped into a cluster around the Drake passage consisting of the SAAMER at Rio Grande, Tierra del Fuego Argentina, King Sejong Station on King George Island and Rothera on the Antarctic Peninsula



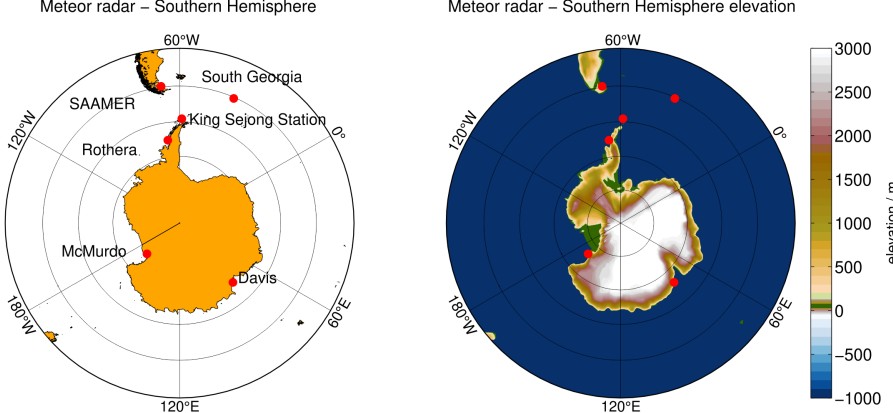

**Figure 1.** Stereographic-projection of the geographic location of the meteor radars used in this study and a map of the terrain elevation of Antarctica, the Antarctic Peninsula and Southern Argentina to visualize the orography around each radar station.

and King Edward Point on KEP. The other two radars are located almost opposite of the Drake passage at McMurdo and Davis Antarctic stations. Figure 2.1 shows two panels with stereographic projections of the Southern hemisphere where the radar locations are represented by red dots (left panel and Table 2.1). The right panel in this figure shows a color contour map of the mean elevations around each radar system to identify potential orographic wave forcing sources underneath the observation volumes.

A technical summary of the radars is provided in Table 2.1. Most of the systems have been in operation for more than a decade and have provided reliable and continuous observations. Although most of these systems have been operated without major parameter changes, both ROT and TDF meteor radars have been upgraded during the observing period of our study. Until February 2019, the ROT system used a high pulse repetition frequency (PRF) meteor mode with a PRF of 2144 Hz, a 2 km range sampling and 4 coherent integrations. After this time, the system was upgraded and resumed operation transmitting a 7-bit Barker code with 1.5 km range sampling and a PRF of 625 Hz. We also noted a significant noise or interference at Rothera before the upgrade in January/February that did not allow to derive trustworthy momentum fluxes. Further, we also restricted our analysis of mean winds and tides to the altitude range between 80-100 km. In addition, in September 2019, the TDF transmitting scheme also changed. The original design of the TDF transmitter (TX) configuration used eight three-element crossed Yagis arranged in a circle of diameter 27.6 m. Each transmitting in opposite phasing of every other Yagi (Janches et al., 2014). In 2019, the system transmission strategy was upgraded with the deployment of a single new TX antenna with the goal of improving the detection rate of meteors at larger zenith angles for astronomical purposes (Janches et al., 2020). By concentrating the full power of TDF in one TX antenna, a more uniform detection pattern is achieved that satisfies this original requirement, but also increases the number of events detected at larger zenith angles. Finally, the MCM radar is the most recent installation which, although it is not the most powerful radar, provides the best altitude coverage. This is partly explained by the sporadic meteor sources and the southern location of the MCM meteor radar. The helion, antihelion and the south apex meteor source are above the local horizon all the time contributing to the observed sporadic meteor fluxes at





**Table 1.** Technical parameters of the meteor radars

|  | TDF | KEP | KSS | ROT | DAV | MCM |
|---|---|---|---|---|---|---|
| Freq. (MHz) | 32.55 | 35.24 | 33.2 | 32.5 | 33.2 | 36.170 |
| Power (kW) | 64 | 6 | 12 | 6 | 7 | 30 |
| PRF (Hz) | 625 | 625 | 440 | 2144/625 | 430 | 500 |
| coherent integration | 1 | 1 | 4 | 4/1 | 4 | 1 |
| pulse code | 7-bit Barker | 7-bit Barker | 4-bit complementary | mono/ Barker | 4-bit complementary | 7-bit Barker |
| sampling (km) | 1.5 | 1.5 | 1.8 | 2/1.5 | 1.8 | 1.5 |
| location (lat,lon) | 53.7°S, 67.7°W | 54.3°S, 35.5°W | 62.2°S, 58.8°W | 67.5°S, 68.0°W | 68.6°S, 78.0°E | 77.8°S, 166.7°E |

MCM, yielding a much weaker seasonality in the altitude variation of the meteor layer. In addition, meteors arriving from
these sources enter the atmosphere at fairly low entry angles (<20° see Schult et al. (2017) for a northern hemisphere radar),
leading to a much smoother ablation profile of the meteoroids and, hence, the released meteoric material is spread over a larger
segment of the meteor flight path increasing the detectability. On the other side, the orbit geometry alone provides not yet a
sufficient explanation for the better altitude coverage at McMurdo, however, a more detailed investigation is beyond the scope
of the paper.  Several of the meteor radars used in this study employ the standard meteor radar configuration an array of 5-Yagi
antennas for reception with a spacing of 2 and 2.5 $\lambda$ (Jacobs and Ralston, 1981; Jones et al., 1998). MCM was set up in a
different configuration with 1.5 and 2 $\lambda$ spacing due to topographic constraints. Similar to other meteor radars most of them
use a single Yagi antenna for transmission. Only TDF employed a beam forming for transmission forming 8 main beams, as
described earlier, but changed to the use of the single crossed Yagi antenna late during the period studied here (Janches et al.,
2014; Janches et al., 2020). A more detailed description of the King Sejong Station meteor radar can be found in Lee et al.
(2018) and for the DAV meteor radar in Holdsworth et al. (2004).

## 2.2 Retrieval of winds and momentum flux

For decades, meteor radars have been used to measure winds in the mesosphere/lower thermosphere (MLT). Typically winds
are obtained by least square fits solving for the horizontal wind velocities after binning the data into altitude and time intervals
(Hocking et al., 2001; Holdsworth et al., 2004). In this study we retrieve winds using the algorithm presented in Stober et al.
(2018a), which includes the treatment of the geometry of the full Earth, based on the WGS84 rotation ellipsoid to provide
more precise altitude estimates and geodetic coordinates for each meteor, a spatio-temporal Laplace filter and a non-linear
error propagation, which is described in more detail in Gudadze et al. (2019). The wind retrievals are cross-validated against
NAVGEM-HA (McCormack et al., 2017; Stober et al., 2019). The results presented in this manuscript are based on winds with





a temporal resolution of 1 hour and a vertical resolution of 2 km. The minimum number of meteors per time and altitude bin

for a successful fit is 4.

Hocking (2005) proposed a method to obtain gravity wave momentum fluxes for typical meteor radars, which was later echoed and reformulated as correlations by Vierinen et al. (2019). In the following, we present a brief derivation of the Reynolds stress tensor and how the different tensor elements are estimated from the meteor radar observations. The starting point is the well-known radial wind equation. Each meteor will form a trail that will be detected by the radar and will drift with

the background wind. The radar will then detect that radial velocity, via Doppler shift in the received signal, and the three components of the background wind can be calculated for each detected meteor using the mathematical convention (reference to east and counterclockwise rotation);

$$v_{\mathrm{rad}} = u \cdot \cos(\phi)\sin(\theta) + v \cdot \sin(\phi)\sin(\theta) + w \cdot \cos(\theta) \quad , \tag{1}$$

where u, v, w are the three wind components (zonal, meridional and vertical, respectively), $\phi$ is the azimuth angle, $\theta$ is the

off-zenith angle and $v_{\mathrm{rad}}$ is the observed radial wind velocity. Further, it is straight forward to use the standard Reynolds decomposition of the wind, separating the wind components into a mean flow $(\overline{u}, \overline{v}, \overline{w})$ and wind fluctuations $(u', v', w')$;

$$
\begin{aligned}
u &= \overline{u} + u' \\
v &= \overline{v} + v' \\
w &= \overline{w} + w' \ .
\end{aligned}
\tag{2}
$$

As we are mainly interested in the momentum flux associated to GWs, the mean flow terms containing the background wind and the diurnal and semidiurnal tide have to be subtracted/removed from the observed radial velocities for each meteor. Thus, we model the mean flow radial velocity by;

$$\overline{v_{\mathrm{radm}}} = \overline{u} \cdot \cos(\phi)\sin(\theta) + \overline{v} \cdot \sin(\phi)\sin(\theta) + \overline{w} \cdot \cos(\theta) \quad , \tag{3}$$

The radial velocity fluctuations $(v'_{\mathrm{rad}})$, which now only contain GW contributions, are obtained by subtracting the mean flow

radial velocity $(\overline{v_{\mathrm{radm}}})$ from the observed radial velocity $(v_{\mathrm{rad}})$ measurements;

$$v'_{\mathrm{rad}} = v_{\mathrm{rad}} - \overline{v_{\mathrm{radm}}} \ . \tag{4}$$

Furthermore, the GW fluctuations can be modelled by;

$$v'_{\mathrm{radm}} = u' \cdot \cos(\phi)\sin(\theta) + v' \cdot \sin(\phi)\sin(\theta) + w' \cdot \cos(\theta) \quad . \tag{5}$$

Considering that these radial velocity fluctuations are mostly driven by GW and, hence, the Reynolds stresses can be computed

by minimizing the following quantity (Hocking, 2005);

$$\Lambda = \sum \left( (v'_{\mathrm{rad}})^2 - (v'_{\mathrm{radm}})^2 \right)^2 \ . \tag{6}$$





Inserting eq. 5 into eq. 6 leads to the well-know momentum flux terms;

$$\Lambda = \sum \Big( (v'_{\mathrm{rad}})^2 - \quad (u'^2 \cdot \cos(\phi)^2 \sin(\theta)^2 + v'^2 \cdot \sin(\phi)^2 \sin(\theta)^2 + w'^2 \cdot \cos(\theta)^2 + 2u'v' \cdot \cos(\phi)\sin(\phi)\sin(\theta)^2 + \tag{7}$$
$$2u'w' \cdot \cos(\phi)\sin(\theta)\cos(\theta) + 2v'w' \cdot \sin(\phi)\sin(\theta)\cos(\theta)) \Big)^2 \quad .$$

Solving eq. 7 for the unknown Reynolds stresses components is straight forward. Typically, the terms $u'w'$, $v'w'$ and $u'v'$ are also called momentum fluxes and the symmetric Reynolds stress tensor is given by;

$$\tau'_{ij} = \rho\overline{u_i u_j} = \rho \cdot \begin{pmatrix} u'^2 & u'v' & u'w' \\ u'v' & v'^2 & v'w' \\ u'w' & v'w' & w'^2 \end{pmatrix} \quad , \tag{8}$$

where $\rho$ is the atmospheric density at the altitude of the measurement and the other terms in the tensor denote the Reynolds stress components (wind variances and momentum fluxes), which have units of squared velocity fluctuations. The Reynolds

stress are derived from the RANS (Reynolds Averaged Navier Stokes) equations assuming an incompressible Newtonian fluid and that the Reynolds average of the fluctuations vanishes ($\overline{u'} = 0$), which requires that the averaging has to be long enough to cover the inertia GW periods of several hours or longer. The spatial scales can theoretically be estimated by selecting different volumes inside the domain area, however, practically the meteor statistics is often not sufficient to get reliable results.

However, there are some caveats of the theory outlined above, when it comes to implement the algorithm and actually to apply

it to meteor radar observations. One difficulty is the Reynolds decomposition into the mean flow and the GW fluctuations. Previous studies often limited the analysis to a narrower angular region (Fritts et al., 2010a; Placke et al., 2015a) using only off-zenith angles between 10-50° reducing significantly the number of meteors for the analysis per time and altitude bin, which in turn required longer averaging or was achieved by an active beam forming antenna. Such an antenna directed more energy towards an angular region as for TDF (Fritts et al., 2010b, a) or the meteor radar at Trondheim (de Wit et al., 2014). The

process by which this much stricter angular selection of meteors improved the momentum flux estimates was the reduction of projection errors due to the Earth's ellipsoid shape, which caused apparent and arbitrary contributions to the fluctuation terms. Stober et al. (2018a) proposed to minimize this type of uncertainty by computing for each meteor its geodetic position relative to the WGS84 reference ellipsoid, which improves the altitude determination, but also reduces projection errors for the azimuth and off-zenith angle. The benefit of this full Earth geometry correction is that there is no longer a need to reduce

the angular region and all meteors up to off-zenith angles of 65° can be used for the analysis. Typical specular meteor radars (single Yagi-antenna on transmission) detect most meteors at off-zenith angles between 50° and 70°. However, meteors at larger zenith angles are further away from the radar and, thus, are more prone to altitude errors. Considering that the typical angular precision of the employed receiver arrays is approximately 1.5-1.7° (Jones et al., 1998). A limit of 65° presents a more optimal choice to maximize the number of meteors entering the analysis while keeping a sufficient altitude precision. Another

important aspect related to the momentum flux estimation is the proper removal of the background flow, which was already outlined by Fritts et al. (2010a); Placke et al. (2011b); Andrioli et al. (2013) and later confirmed by de Wit et al. (2014). In particular, tides have large amplitudes in the MLT, causing large vertical and temporal shears within a time and altitude bin.





Noting that Hocking (2005) and Placke et al. (2011b) suggested the use of at least 30 meteors for a successful momentum flux fit, which is often achieved by temporal averaging, the importance of the temporal shear becomes evident.

In this study we use the adaptive spectral filter (ASF) to perform the Reynolds decomposition into a background flow and the GW fluctuations . A first version of the ASF(1D) (temporal domain) was presented in Stober et al. (2017). Meanwhile a ASF(2D) is developed, which employs a vertical regularization constraint for the mean wind and tides assuming a smooth vertical phase progression for each wave without an explicit vertical wavelength threshold (Pokhotelov et al., 2018; Baumgarten and Stober, 2019; Wilhelm et al., 2019). The ASF accounts for the continuous variation of the mean flow as well as for the

intermittent behaviour of the tides. Thus, we obtain hourly resolved background wind fields for each altitude and time bin for the zonal, meridional and vertical wind component, respectively. This background wind field contains the mean flow and the diurnal, semidiurnal and terdiurnal tidal component. However, the terdiurnal tide has usually a much smaller amplitudes compared to the other tides and is therefore not further discussed. Further, we perform a linear interpolation of this background wind field to the actual occurrence time and altitude for each meteor to estimate the $v_{\mathrm{radm}}$-term minimizing any contribution

from the background flow. This procedure is effective in mitigating possible contamination due to tides, and permits to use of much longer averaging windows. In this study we use 64-hours and a minimum of 100 meteors to determine the Reynolds stress components. However, for the seasonal climatology only solutions with more than 1000 meteors enter the statistics.

The algorithm is implemented similar to that performed for wind retrievals in Stober et al. (2018a). The first guess is provided by a classical least square fit. Based on this initial iteration, we compute the spatio-temporal Laplace filter, which

provides a predictor for each time and altitude bin. This predictor enters as regularization (Tikhonov) all further iterations and is updated each time. The spatio-temporal Laplace filter turns out to be beneficial for ill-conditioned problems due to the random occurrence of meteor detections and asymmetries in the spatial sampling; these can result in difficulties in determining all parameters with similar quality.

Furthermore, we perform a non-linear error propagation similar to the one presented in Gudadze et al. (2019). The statistical

uncertainties are updated in each iteration step. We also tested barrier functions to penalize negative values of $u'^2, v'^2$ and $w'^2$, respectively. Such negative values were reported in Placke et al. (2011a), but apparently this is a minor issue in our retrievals. Only a negligible number of fits resulted in negative values for just some of the radar systems utilized in this work.

In addition, we performed some test retrievals to account for the vertical velocity bias intrinsic to the meteor radar observations. Specular meteors have trail lengths of up to several kilometers where the radio waves are scattered and, thus, meteors

entering the Earth's atmosphere at steep entry angles can encounter strong vertical wind shears, which lead to a rotation of the trail causing systematic errors. In particular, during the local summer months this can lead to a systematic deviation of a few cm/s for mid-latitude stations.

Very often wind fits are performed by assuming $w = 0$ m/s (Hocking et al., 2001; Holdsworth et al., 2004). However, here we use the retrievals as presented in Stober et al. (2018a) who used the vertical wind velocity as quality control. Typically,

we obtain daily mean values of the order of $\pm 0.25$ m/s, which is more than an order of magnitude less than reported by Egito et al. (2016). However, the remaining bias due the vertical winds, which potentially has the wrong sign, had no impact on the retrieved Reynolds stresses. Finally, in order to get confidence in the retrievals, we performed several test cases similar to the


ones presented in Fritts et al. (2010a). Therefore, we extracted the observed meteor detections from TDF and synthesized wind
fields including altitude dependent mean winds and tides with a vertical wavelength of 80 km and various altitude dependent
GW fields to optimize the retrieval setting with respect to the regularization strength, the required statistics and the applied
averaging. Performing these tests we find minor deviations from the synthetic wind and GW fields only at the upper and lower
edges of the meteor layer. The tidal amplitudes were retrieved within $\pm 2$ m/s compared to the synthetic data. The momentum
fluxes agreed for the 30-day median remarkably well. We also tested the possibility to retrieve the vertical wind fluctuation
amplitudes and found mean deviations of $\pm 0.01$ m/s residual bias for the synthetic fields and about $\pm 0.25$ m/s bias in our
observations.

### 2.3   MLS satellite observations

To bring the local radar observations into the global context we calculate the geostrophic zonal wind as described in Matthias
and Ern (2018) from geopotential height (GPH) data from the Microwave Limb Sounder (MLS) on board the Aura satellite
(Waters et al., 2006; Livesey et al., 2015). MLS has a global coverage from $82°$S to $82°$N on each orbit, and a usable height
range from approximately $11$ to $97\,km$ $(261-0.001\,hPa)$ with a vertical resolution of $\sim 4\,km$ in the stratosphere and $\sim 14\,km$
at the mesopause. The temporal resolution is 1 day at each location, and data are available from August 2004 until the present
(Livesey et al., 2015). Version 4 MLS data were used in this paper along with the application of the most recent recommended
quality screening procedures from Livesey et al. (2015). For our analyses the original orbital MLS data are accumulated in grid
boxes with $10°$ grid spacing in longitude and $5°$ in latitude. Afterwards they are averaged at every grid box and for every day,
generally resulting in a global grid with values at every grid point.

### 3   Results

#### 3.1   Mean winds 2019

As pointed out in the previous section, we have to perform a Reynolds decomposition separating a mean flow from the GW
fluctuations. Thus, we analyze the data with the adaptive spectral filter (ASF) technique (Baumgarten and Stober, 2019) to
obtain daily mean winds, as well as diurnal and semidiurnal tides.
We first compare the seasonal zonal and meridional winds of all 6 locations to identify any seasonal and local differences. Fig-
ure 2 shows the seasonal zonal and meridional wind pattern during 2019 obtained from the daily mean zonal and meridional
winds after applying a 30-day running median shifted by one day. This reveals any seasonal variability by removing atmo-
spheric waves with shorter periods. A similar analysis was applied in Wilhelm et al. (2019) for meteor radars in the Northern
hemisphere in order to derive mean wind climatologies. Significant differences between the locations can be observed from
this figure, in particular during the southern hemisphere winter seasons (JJA). Both, TDF and KEP observations, performed at
almost the same latitude at $54°$S, show a similar morphology for the zonal winds and only the meridional winds deviate during
June and July from each other. At the tip of South America, TDF shows that the meridional winds experience a sign reversal





**Figure 2.** Comparison of zonal and meridional mean winds for each station during the year 2019 for a) TDF, b) KEP, c) KSS, d) ROT, e) DAV and f) MCM.



around June/July, which is not present over KEP. The meridional winds seem to also have a semiannual oscillation at both
locations.

Further polewards at KSS and ROT (62°S and 67°S, respectively), which are at a similar longitude as TDF, the zonal winds
reflect a similar seasonal behaviour compared to KEP and TDF, but with a slightly weaker wind magnitude. However, the
meridional winds are fairly consistent for the summer months compared to the mid-latitude radars, but deviate considerably
for the winter season. There is even a noticeable difference between KSS and ROT, even though the systems are located rather
close together. At ROT and KSS the meridional winds show only during April, May and June a typical winter behaviour and
approximately northward winds for the other months above 80-85 km. Only during September and at altitudes above 90 km
above KSS a short southward wind patch occurs.

Comparing the observed wind fields measured at ROT and DAV, which are only separated by 2 degrees in latitude, but by
170° in longitude underlines even more clear that there is a significant asymmetry in the southern hemispheric wind systems.
As expected, looking at the general morphology, the seasonal zonal wind pattern for 2019 is remarkably similar between both
locations. There are only marginal differences in the zonal magnitudes considering the overall agreement of the zonal wind
structures. This is also the case for the meridional winds during the summer months (DJF). However, during the winter sea-
son the meridional wind structure is significantly different between both stations. The morphology at DAV appears to be less
asymmetric with a tendency to show increased southward wind magnitudes towards the end of the winter season, whereas at
ROT the highest southward winds are registered at the begin of the winter season 2019.

The southernmost location in our analysis is MCM at 78°S. The seasonal zonal wind morphology compares well with those
measured at DAV and ROT, but shows much weaker wind magnitudes. Similar to observations in the Northern hemisphere the
summer zonal wind reversal altitude also increases with increasing southern latitude. Compared to DAV, the meridional winds
are intensified during the summer and winter seasons. Furthermore, the asymmetry during the winter months is also present
at MCM, which shows, similarly to DAV, the highest southward meridional winds towards the end of the winter season as a
double structure. In fact, the southward meridional winds at MCM during July and September 2019 are the strongest of all
locations.

## 3.2 Diurnal tidal amplitudes and phases measured during 2019

Atmospheric tides provide a time variable background filter for the vertical propagation of GWs, which can, depending on
the tidal phase and the propagation direction of the GW, lead to GW breaking and dissipation. These breaking events might
trigger/foster the generation of secondary or non-primary waves (Heale et al., 2020). Thus, tides are essentially contributing
to the Reynolds decomposition. In particular, the day-to-day variability is crucial for the momentum flux analysis. Typically,
atmospheric tides are derived assuming phase stability over a certain period of time, which can be several days, weeks or
months (Murphy et al., 2006; Hoffmann et al., 2010; Conte et al., 2017; He et al., 2018; Pancheva et al., 2020). More recent
studies favor much shorter windows of 24 to 48 hours to account for the intermittent behaviour of tides (Stober et al., 2017;
Wu et al., 2019; de Araújo et al., 2020; Das et al., 2020), in particular, phases of atmospheric tides that appear not to be
constant with time (Ward et al., 2010; Baumgarten and Stober, 2019; Stober et al., 2019). In this study, all tidal amplitudes and





**Figure 3.** Same as Figure 2 but for the diurnal tidal amplitudes.





**Figure 4.** Same as Figure 2 but for the diurnal tidal phases.



phases were determined with the ASF which, similar to wavelet spectra, adapts the window length to the period of the fitted frequencies. The obtained daily tidal amplitudes are vector averaged using 30-day medians centered at the respective day to

derive the seasonal variation.

Figure 3 presents the seasonal variation of the diurnal tidal amplitudes measured at each station. Although the daily mean winds showed significant differences between TDF, KEP, KSS and ROT, the seasonal behaviour of the diurnal tide is rather consistent between all four locations. There is a pronounced summer maximum in the zonal and meridional amplitudes from January to February at altitudes from 78-106 km. The meridional tidal amplitudes tend to exceed the zonal amplitude by up to

10 m/s. At altitudes above 100 km the diurnal tide remained of significant magnitude until May 2019. Apparently, there are no significant diurnal tidal amplitudes (< 10 m/s) visible at altitudes between 80-100 km for the rest of the year 2019. KSS and ROT indicate a small diurnal tidal enhancement for July/August and in December below 80 km and above 100 km altitude. The December enhancements are also found at TDF, but almost disappear at KEP. DAV measurements show basically the same seasonal diurnal tide behaviour, but with weaker amplitudes. The winter diurnal tidal enhancement in June/July appears to be

more pronounced. However, the southernmost meteor radar at MCM observes a significantly different seasonal diurnal tidal pattern. The summer maximum is much more pronounced compared to the other stations and shows amplitudes of 20 m/s from January to April at 90 km and above and again from October to December. There is also a noticeable difference between the zonal and the meridional diurnal tidal amplitude. The zonal component indicates a winter minimum, whereas the meridional component shows a tidal enhancement.

Diurnal tidal phases are shown in Figure 4. The tidal phases are given in UTC, hence, longitudinal differences are present as phase shifts. As expected the diurnal phases are much more variable during time with low tidal amplitudes for TDF, KEP, KSS and ROT. During the summer months of January and February 2019 the diurnal phases are more stable and indicate rather long vertical wavelengths, but with significant differences between the zonal and the meridional components. However, the phase plots indicate a distinct seasonal pattern showing phase drifts of several hours at the same altitude over the course of the

year. The more south the meteor radar is located the less characteristic is the seasonal behaviour. DAV and MCM indicate a decreased variability of the diurnal tidal phases throughout the year. At DAV there are even times visible, which suggest almost phase stability over several weeks, instead of the typical continuous variation reflected by the other stations.

### 3.3 Semidiurnal tidal amplitudes and phases measured during 2019

At mid- and high latitudes, semidiurnal tides are the dominating tidal wave during the course of the year (Hagan and Forbes,

2002, 2003). Figure 5 shows the vector averaged semidiurnal tidal amplitudes measured by all six meteor radars using again a 30-day median shifted by one day in analogy to the mean winds and diurnal tides.

The seasonal structure of the semidiurnal tide reveals a rather interesting pattern for the southern hemisphere. Semidiurnal tides measured at TDF, KEP, KSS and ROT show some similarities for the zonal tidal component, resulting in amplitudes below < 10 m/s for the summer months January to mid-March. From April to June all 4 stations show a strong semidiurnal tidal activ-

ity with amplitudes up to 40 m/s, another minimum of the tidal activity in July and a secondary maximum from August to the end of the year. Furthermore, the tidal amplitudes show a decrease with increasing polar latitude, which is also observed at the





a) 

b) 

c) 

d) 

e) 

f) 

**Figure 5.** Same as Figure 2 but for the semidiurnal tidal amplitudes.





**Figure 6.** Same as Figure 2 but for the semidiurnal tidal phases.



**Figure 7.** Same as Figure 2 but for the semidiurnal tidal vertical wavelengths.





northern hemisphere. However, the meridional semidiurnal tide shows a clear longitude dependence and asymmetry compared to the zonal tidal amplitudes, which was not reported previously (Conte et al., 2017). At the longitude of TDF and ROT the meridional tidal component is much weaker during April to June compared to the zonal. At KSS, which is more towards the

East, similar amplitudes for the zonal and meridional component are observed. This was also found at KSS in a previous study analyzing the tidal amplitudes under solar maximum conditions, which resulted in larger amplitudes of the semidiurnal tide (Lee et al., 2013). At KEP, which is 25° eastward shows the opposite behaviour and the meridional semidiurnal tide reaches the highest amplitudes in April to June.

The semidiurnal tidal seasonal behaviour at DAV looks quite different from the stations that are located further to the north.

The amplitudes are much weaker and barely reach values of 25 m/s and there are 4 periods with an increased activity, which are during January-February, May, August-September and in December. The largest tidal amplitudes are observed in May 2019.

Further to the south, at MCM Station, the semidiurnal tide exhibits only a very faint seasonal structure. Most of the time the amplitudes are below 10 m/s. Only during March, May and November-December and below 90 km altitude there are times where amplitudes exceed 10 m/s. This is surprising when we compare these values with the geographically conjugate northern

hemisphere latitude. At Svalbard (78.17° N, 15.99° E) the semidiurnal tide still reflects a similar seasonal activity as other polar and mid-latitude locations (Wilhelm et al., 2019; Pancheva et al., 2020). This is obviously not the case in the southern hemisphere and represents a remarkable interhemispheric difference.

Semidiurnal tidal phases are displayed in Figure 6 where it can be seen that the semidiurnal tidal phases reflect similar features than those present in the amplitudes. TDF, KEP, KSS and ROT show a very similar seasonal structure indicating continuous

changes of the tidal phases throughout 2019 at all altitudes. At DAV and MCM, on the other hand, the phases indicate an even more pronounced seasonal structure and faster gradual phase drifts. In particular, at MCM the phases appear to be more variable, which is likely due to the generally weaker amplitudes pointing towards a much weaker and more intermittent excitation of the tides. Comparing the seasonal phase behaviour of the southern hemisphere to conjugate latitudes in the northern hemisphere points out that there are some differences. In the northern hemisphere, from mid- to high latitudes, the semidiurnal tides

show a seasonal asymmetry between the winter to summer transition and the fall transition. In the northern hemisphere the fall transition is accompanied by a significant phase change from September to November, whereas in the southern hemisphere this feature is very weak at TDF and ROT (March to May), and almost negligible for KEP and KSS.

Finally, we briefly discuss the presence of a potential lunar tide. Sandford et al. (2006, 2007) estimated the lunar tide amplitude from two northern hemispheric meteor radars and Davis MF-radar in the southern hemisphere and found values of 1-2 m/s,

which is negligible compared to the typical GW amplitudes of about 20-30 m/s for the resolved waves. However, Forbes and Zhang (2012) investigated a potential lunar tide amplification due to the Pekeris resonance. They found favorable conditions to shift the Pekeris peak towards the lunar tide periods M2 (12.42 h) and N2 (12.66 h), during the time of major sudden stratospheric warming in 2009 in the northern hemisphere, as only during the time of the wind reversal the vertical temperature and wind structure satisfies the resonance condition. Later, Zhang and Forbes (2014) argued that the Pekeris resonance peak is

rather broad and, thus, more or less each sudden stratospheric warming can cause a lunar tide amplification.

These reports triggered several studies investigating the lunar tide and its relevance for the mesosphere dynamics. However,





most of the observational diagnostics (Wavelet or harmonic fitting) separating the lunar tide from the semidiurnal tide applied long windows of 21-days or even longer periods up to several months assuming phase stability of the semidiurnal tide (Forbes and Zhang, 2012; Chau et al., 2015; Conte et al., 2017; He et al., 2018; Siddiqui et al., 2018). However, as shown in Figure 6,

the semidiurnal tidal phase shows considerable variability and seasonal changes and, thus, the assumption of phase stability for the semidiurnal tide is not valid. Therefore, we performed a holographic analysis to test whether a temporally variable semidiurnal tidal phase could be misinterpreted as a lunar tide (see appendix A1) (Stober et al., 2019) . In fact, the holograms often exhibit a shift towards the M2 frequency (12.42 h) uncorrelated with the lunar orbit. Given these results and considering that there was only a minor stratospheric warming in September 2019 (Yamazaki et al., 2020), we consider the lunar tide as a

minor wave with a negligible amplitude compared to GWs and did not make an attempt to remove this tidal component in our Reynolds decomposition.

For the sake of completeness, we also estimated the vertical wavelengths of the semidiurnal tide, which is presented in Figure 7. The vertical wavelength provides a good overview to identify potential changes in the Hough modes of the tide. The vertical wavelengths show a similar latitude and longitude dependence as already discussed for the semidiurnal tidal ampli-

tudes and phases. The observations at TDF, KEP, KSS and ROT indicate almost the same vertical wavelengths from March to October 2019 of about 70-100 km. This corresponds to the time with the largest semidiurnal tidal amplitudes. However, the seasonal summer months January-February and November-December show a longitudinal difference. KEP and KSS observe much longer vertical wavelengths of up to 1000 km during these months, compared to the stations located to the west. These very long vertical wavelengths are associated to times with a small semidiurnal tidal amplitude. The results obtained at DAV

reflects a slightly different seasonal behaviour. There, the longest vertical wavelengths are observed in March-April followed by a stable hemispheric winter season until August and a gradual decreasing vertical wavelengths towards the end of the year. The results at MCM show an even more complicated picture due to the almost vanishing semidiurnal tidal amplitudes. Only during the local summer months of January/February and November/December are meaningful vertical wavelengths derivable, with vertical wavelengths of about 70-100 km. It is also worth mentioning that the agreement between the zonal and meridional

wavelengths is remarkable and provides further confidence in the applied ASF technique used for the Reynolds decomposition.

### 3.4 Reynolds stress components

Gravity waves are an essential driver of the MLT dynamics and variability carrying energy and momentum from their source region to the altitude of their deposition. The breaking of GWs can trigger the generation of non-primary gravity waves, which again can propagate upwards (Becker and Vadas, 2018; Vadas and Becker, 2018) causing a complex interaction chain for the

GW activity and the resulting forcing at the MLT. The acceleration/deceleration of the mean flow due to momentum and energy transfer by breaking GW can be estimated from the vertical gradient of gravity wave momentum flux (Ern et al., 2011).

From our Reynolds decomposition and the retrieval we determine three momentum fluxes, which are often referred to as the vertical flux of zonal momentum $< uw >$, the vertical flux of meridional momentum $< vw >$ and the horizontal momentum flux $< uv >$, where the $<>$ denotes temporal averaging.

Figure 8 shows all three momentum flux components as a 30-day median shifted by one day for the year 2019. There are



**Figure 8.** Comparison of vertical flux of zonal and meridional momentum and horizontal momentum flux for each station during the year 2019 for a) TDF, b) KEP, c) KSS, d) ROT, e) DAV and f) MCM.

**Figure 9.** Same as Figure 8, but for the zonal, meridional and vertical wind variances.





three panels for each station presenting the vertical flux of zonal momentum, the vertical flux of meridional momentum and the horizontal momentum flux. The results shown in this figure indicate that, for all six meteor radar observations, there is a characteristic seasonal pattern with noticeable differences between the different locations. Only the horizontal momentum flux $< uv >$ shows a similar seasonal behaviour at TDF, KEP, KSS, ROT and DAV with negative values of -50 $m^2/s^2$ during the

seasonal summer and positive values in winter from April to October. The local winter shows more variability and a semiannual structure for some sites, similar to the mean zonal and meridional winds. At MCM this seasonal variation is still visible, but with a much weaker magnitude.

The vertical flux of zonal momentum $< uw >$ is rather variable with longitude and latitude. Observations at TDF and ROT show some similarities regarding the seasonal structure. During the local summer both indicate positive zonal momentum

fluxes at the altitude of the zonal wind reversal. At higher altitudes, above 95-100 km, the zonal momentum flux reverses to negative values. The winter season appears to be more variable, which might be related to the minor warming in September 2019 and the wave activity before. To the east, at KEP and KSS, positive zonal momentum fluxes at the higher altitudes (95-105 km) are observed throughout the year, but a rather different behaviour during the local winter season at the altitudes below. In Particular, at KSS a variable zonal momentum flux is measured that seems to be in better agreement with TDF and ROT results.

Further to the South, at DAV and MCM, the seasonal behaviour of the zonal momentum flux seems to reflect the features that are already found at KEP, but with different magnitudes.

The vertical flux of horizontal momentum presented in Figure 8 exhibits some longitudinal dependence. Observations at TDF and KEP show approximately the opposite vertical structure of the meridional momentum flux pointing out that the meridional drag (acceleration/deceleration) reverses between their longitudes. Furthermore, results from KSS and KEP show a good

agreement of the vertical structure and seasonality of the meridional momentum flux throughout the year. Results from ROT and DAV still reflect some features of the seasonal meridional momentum flux behaviour, but with decreasing magnitude, while at MCM, the results show once again a seasonal dependency comparable to those obtained at KEP.

The Reynolds stresses on the main diagonal of the Reynolds stress tensor are also investigated for their seasonal dependencies. These terms are also often called zonal, meridional and vertical wind variances. Figure 9 shows all three variances for each

station. The color scale for the vertical variances is 5 times smaller compared to the horizontal wind fluctuations. The zonal and meridional variances exhibit a seasonal structure and a rather obvious altitude dependence. The highest variances are observed at the highest altitudes, which is expected considering that the Reynolds stresses are weighted by density exponentially decreasing with altitude. It is also a common feature for all sites that the meridional velocity variances exceed the zonal fluctuations. The seasonal behaviour of the zonal and meridional variances at all stations reflects a semi-annual variation

showing minimum variances during the equinoxes, when the mean winds are smallest at altitudes below 95-100 km. Above 100 km the seasonal characteristic appears to be less pronounced. The vertical wind variances are the most challenging values to retrieve. Their seasonal behaviour is less obvious. However, the vertical wind variances also indicate increasing values with decreasing density. Results at MCM are exceptional in this respect and the vertical wind variances exceed the values that are derived at all other stations. At present we can only speculate on the source of these large values. A possibility is that MCM

lies underneath the auroral oval and, thus, the altitudes above 90 km are strongly influenced by precipitating particles and





associated effects like Joule heating that might trigger stronger vertical variations.

To gain confidence in our retrieved horizontal wind variances, we performed a test by estimating the GW wind variances of the resolved GW waves directly. It is straightforward to derive a gravity wave residual from the hourly observed wind time series by subtracting mean winds and the diurnal and semidiurnal tide. Thus, we obtain a hourly time series of the GW residuals,

which corresponds to the kinetic energy of the resolved GW waves with periods longer than 2 hours and horizontal wavelengths of more than 300 km, whereas the wind variances obtained using Hocking (2005) include the GW variances from all temporal and spatial scales. The GW variances from the residuals are shown in Figure A2.

## 4 Discussion

Meteor radar observations of gravity wave momentum fluxes have been performed for more than a decade (Hocking, 2005;

Fritts et al., 2010a; Placke et al., 2011b; Andrioli et al., 2013; de Wit et al., 2014, 2017). However, the results were not always conclusive and often difficult to interpret. Many of these former studies focus on understanding the method and how to optimize the analysis procedure (Fritts et al., 2010a; Placke et al., 2011b; Andrioli et al., 2013; Placke et al., 2015a). Although the Reynolds decomposition appears to be rather straight forward, it can be quite challenging to do a proper and robust implementation to separate the mean flow from the gravity wave fluctuations. Fourier based methods often require long averaging

windows to get a proper resolution, but do not capture sufficiently the intermittency of the background (see Figure A3). For shorter windows the irregular sampling of meteors in time and altitude again causes deviations from regular grids and data gaps have to be considered when applying wavelet or Fourier techniques. Another complication of the meteor radar momentum fluxes is that there are no 'ground truth data' available to validate the measurements. Satellite observations provide only a total gravity wave momentum flux without directional information obtained from temperature fluctuations after removing

atmospheric tides up to wave number 4 assuming a stationary phase behaviour over a couple of days (Ern et al., 2011; Trinh et al., 2018), and thus confidence on the methodology relies on test with synthetic fields such as those presented in Fritts et al. (2010a) and Fritts et al. (2012a).

As we are mainly interested in the GW momentum flux and wind variances, we have to evaluate carefully the presence of potential error sources in the Reynolds decomposition methodology. In particular, atmospheric tides show a very intermittent

behaviour of the amplitude and phase, which causes some issues in the decomposition when long windows (several days/weeks or months) are used. Thus, we performed some tests to optimize the mean flow, tidal and gravity wave decomposition applying the ASF, a 1-day harmonic fit and a 5-day harmonic fit (see appendix A3). The comparison indicates that the Reynolds decomposition tends to be very sensitive to the applied technique impacting the tidal mean flow and the gravity wave variances. Hence, the derived momentum fluxes and wind variances can be significantly different, even though the same or similar data

sets are used. Previous studies used 4-day fits (de Wit et al., 2014) or S-transforms (de Wit et al., 2017) to decompose the time series. In fact, the harmonic fits provide similar results compared to those obtained using Fourier based techniques such as the S-transform (Stockwell et al., 1996) or wavelets (Torrence and Compo, 1998) for the same averaging length.

Besides these technical aspects of the momentum flux and wind variance retrievals, the year 2019 was exceptional in the south-





**Figure 10.** Monthly mean geostrophic zonal wind for the austral winter 2019 (a, b, c) and summer 2018/19 (d, e, f) each averaged over the altitude range of its wind maximum (40-60 km for winter, 60-80 km for summer). Black dots mark the positions of the radar stations used here. Data are derived from MLS geopotential height data.



ern hemisphere (SH). The SH winter season was much more variable in August/September compared to previous years and

disturbed with a rare minor sudden stratospheric warming occurring in September (Yamazaki et al., 2020). This variability is

reflected in the mean winds and the momentum fluxes, which show noticeable longitudinal and latitudinal differences pointing

towards an unstable and wobbling polar vortex. Figure 10 presents monthly mean geostrophic winds from MLS (Matthias and

Ern, 2018) averaged over the altitude range 40-60 km for winter months (a, b, c) and 60-80 km for summer months (d, e, f), the

maximum wind region in each season (not shown). During winter the polar vortex is characterized by a strong longitudinal and

latitudinal variation. The strength of the polar vortex appears to be rather different with longitude and month providing signifi-

cant differences for the vertical propagation of gravity waves and their encounters with critical lines, fostering wave breaking

and the emission of non-primary waves due to localised body forces (Vadas and Fritts, 2001; Becker and Vadas, 2018; Vadas

et al., 2018; Dong et al., 2020; Fritts et al., 2020).

This polar vortex wobbling is essentially modifying the mesospheric gravity wave activity and the resulting momentum flux at

the altitude of the wave breaking in the mesosphere and above. Characteristics of GWs in the MLT strongly depend on their

vertical propagation path and the background wind field along this path, which efficiently alters the amplitude growth of the

gravity waves depending on their phase velocity relative to the mean flow. On the other hand, breaking gravity waves deposit

momentum on the mean flow and, thus, enhance/weaken/shift the polar vortex, contributing further to the wobbling especially

in the mesosphere. For example, the wind maximum in the meteor radar zonal winds (see Fig. 2) was at lower altitudes in June

but at upper altitudes in July and August for most of the stations. One explanation of this phenomenon could be that the polar

vortex in the upper stratosphere/lower mesosphere (USLM, see Fig. 10) was relatively weak in June for most of the stations

while is was considerable stronger especially in July and August. This hypothesis is confirmed by the observations obtained

at KEP, where stronger winds are observed in the USLM already in June resulting in a higher wind maximum compared to

the other stations. Furthermore, there are considerable longitudinal differences of orographic gravity wave sources in the SH

resulting already in some asymmetry at stratospheric altitudes.

During the summer months the differences in the zonal wind between the different stations in the mesosphere are much smaller

than in winter (see Fig. 10) resulting in smaller differences between the stations in the meteor radar observation window (see

Fig. 2).

In particular, the SH winter season shows remarkable longitudinal and latitudinal changes in the strength of the zonal wind

velocities (see Figure 10). Observations with TDF and the KEP radar, although at the same latitude, show that the polar vortex

is rather different at each longitude significantly modifying the conditions for vertical GW propagation, and, thus triggering

differences in the altitude of the momentum flux deposition. Furthermore, it is likely that both stations see different gravity

wave sources. ROT and KSS are also reflecting a remarkable difference in the momentum fluxes, which are partly explainable

by the mesospheric zonal wind field, which shows a rather strong gradient above the Antarctic Peninsula leading to differences

of the polar vortex above KSS and ROT.

The most consistent results are obtained for the horizontal momentum fluxes $< uv >$ and the wind variances. TDF, KEP, KSS,

ROT and DAV observe a very similar seasonal behaviour and only the strength of the flux differs between the sites. This is also

the case for the wind variances, which are very consistent between the meteor radars providing confidence in the retrievals.





The orography around the meteor radars plays only an indirect role in the observed mesospheric momentum fluxes and wind
variances. The observed total flux above the station is the result of all gravity waves that propagate into the mesosphere inde-
pendent of their origin (e.g. jet instabilities, convection, orography or non-primary waves). Satellite observations of the total
momentum flux showed at the stratosphere a significant GW hot spot around the Andes and Antarctic Peninsula (Ern et al.,
2011). However, with increasing altitude this momentum flux forms a plume stretching downwind of the Andes and Antar-
tic Peninsula, and, at the mesosphere/lower thermosphere one finds more or less a longitudinal band of the momentum flux
confined to the latitudinal band between 40°-65°S (Trinh et al., 2018). GCM modelling with a GW resolving model indicated
that the zonal and meridional momentum flux shows a latitudinal and longitudinal structure Becker and Vadas (2018) with
multiple sign reversals of the momentum flux within this latitude band. In particular, the signs of the resulting momentum flux
are opposite between the Antarctic Peninsula and the Andes around TDF. Furthermore, the momentum fluxes exhibit a reversal
of the GW drag between TDF and KEP at the mesosphere, although both locations are at the same latitude.

## 5  Conclusions

This study presents an overview of gravity wave momentum fluxes and wind variances at the MLT in the southern hemisphere
from the mid-latitudes at TDF, Argentina and KEP to the polar latitude of DAV and MCM Antarctic Stations as well as King
Sejong Station and ROT at the Antarctic Peninsula for the year 2019. The year 2019 was exceptional and, in particular the
hemispheric winter season appears to be more disturbed than previous years resulting in a rare minor stratospheric warming in
September.
We briefly summarized the derivation of GW momentum fluxes and wind variances derived from meteor radar observations ap-
plying a Reynolds decomposition to obtain the Reynolds stress components for the Reynolds stress tensor. A similar approach
was presented in Hocking (2005), who introduced the momentum flux and wind variances analysis based on matrix inversion.
Here we present a new retrieval algorithm which includes a full earth geometry treatment, full non-linear error propagation and
a spatio-temporal Laplace filter as a regularization constraint. The full earth geometry, based on the WGS84 reference ellip-
soid, reduces the altitude uncertainty for each observed meteor and, thus, permits the inclusion of meteors at lower off-zenith
angles down to 65°, instead of the previously used limit of 50° (Fritts et al., 2010a; Placke et al., 2011b; de Wit et al., 2017).
Depending on the transmit and receive antenna beam pattern, this increases significantly the number of meteors, making this
retrieval applicable to less powerful meteor radars compared to the state of the art momentum flux systems such as TDF (Fritts
et al., 2010b), DrAMMER (Fritts et al., 2012b) and the Trondheim meteor radar (de Wit et al., 2014). Fritts et al. (2012a)
showed that conventional meteor radars having lower power and a single transmitting antenna were able to provide reasonable
estimates of GW momentum fluxes at the altitudes having the highest meteor counts. However, the estimates derived were
subject to uncertainties of ∼20 to 50% rapidly becoming excessive at higher and lower altitudes. Nevertheless, the method
presented here enables the calculation of momentum fluxes also for standard meteor radars compensating partly the effects
mentioned above using the spatio-temporal regularization constraint.
The Reynolds decomposition of the wind from meteor radar observations makes an essential contribution to retrieved mo-





mentum fluxes and wind variances. All results shown in this study are obtained by using the ASF to separate the mean flow, containing a mean background wind and the diurnal and semidiurnal tides, from the gravity wave fluctuations. The benefit of the ASF is given in the combination of temporal and spatial (vertical) information of the tides and mean winds to derive the

GW fluctuations. The sensitivity of the Reynolds decomposition on the method to estimate the GW fluctuation was outlined by comparing the ASF, to a 1-day (24-hour) harmonic fit and a 5-day harmonic fit, which showed significant differences in the resulting GW residuals. Thus, comparisons to other momentum flux analysis are rather challenging as potential differences are easily explainable by the applied method, which is often not described in sufficient detail.

We also performed a detailed analysis of the mean zonal and meridional wind for the year 2019 to explore the longitudinal

and latitudinal differences. We noticed significant differences on relatively small regional scales, for instance between KSS and ROT. In addition, we found a strong dependence of the zonal and meridional wind pattern during the southern hemispheric winter season indicating an asymmetric structure of the polar vortex at the MLT. This asymmetry was verified by MLS geostrophic zonal wind observations at the stratosphere and mesosphere, which revealed longitudinal differences of the intensity as well as an altitude dependence of the polar vortex leading to temporal and spatially variable filter conditions for the vertical propaga-

tion of gravity waves.

Furthermore, we investigated the diurnal and semidiurnal tidal seasonal variation of the amplitudes and phases for all six stations to assess longitudinal and latitude differences similar to the mean winds. The diurnal tide showed a rather consistent behaviour of the amplitude and phases measured at TDF, KEP, KSS, ROT and DAV and entirely different seasonal response over MCM. Diurnal tidal phases appeared to be most variable during the local winter season, where the smallest amplitudes

are observed. Semidiurnal tides indicated a more complex seasonal structure exhibiting a rather strong difference in amplitude between the zonal and meridional component for TDF, KEP and ROT, which was not reflected by the semidiurnal tidal phases. In addition, with increasing southern latitudes the amplitude of the semidiurnal tide decreases and shows a different seasonal structure at DAV and basically vanishes over MCM. We also investigated the seasonal characteristic of vertical wavelengths. For TDF, KEP, KSS and ROT, we found vertical wavelengths between 80-100 km for the hemispheric winter season from

mid-February to October 2019. Moreover, there was a tendency of increased vertical wavelengths during times with very small semidiurnal tidal amplitudes.

The vertical fluxes of zonal and meridional momentum show also distinct latitudinal and longitudinal variations, which are expected considering the wobbling of the polar vortex, which is likely the result of longitudinal and latitudinal variations of GW sources e.g., orography and convective or dynamic instabilities. These waves encounter location dependent critical line

filtering due to the structure of the polar vortex and, thus, deposit their energy and momentum at different altitudes and may launch non-primary gravity waves due to localised body forces (Vadas and Fritts, 2001; Becker and Vadas, 2018; Vadas and Becker, 2018; Vadas et al., 2018; Heale et al., 2020), further enhancing the wobbling of the polar vortex with increasing altitudes. However, our results also indicate a remarkable agreement of the horizontal momentum fluxes reflecting a seasonal behaviour with opposite sign from summer to winter. Moreover, the horizontal momentum flux indicates more variability dur-

ing the southern hemispheric winter months.

Finally, we retrieved the wind variances at all stations. These wind variances exhibit a seasonal behaviour with minimum vari-





ances during the equinoxes. In general, the meridional wind variances exceed the zonal components. Besides some differences in the absolute values of the wind variances all observations feature remarkable similarities throughout the year 2019.

*Data availability.* The meteor radar data used in this study from ROT and KEP is from Mitchell, N. (2019): University of Bath Skiymet me-
teor radar data collection. Centre for Environmental Data Analysis, 2020. https://catalogue.ceda.ac.uk/uuid/836daab8d626442ea9b8d0474125a446. The DAV radar data are available upon request from Damian Murphy (damian.murphy@aad.gov.au). They are described at https://data.aad. gov.au/metadata/records/Davis_33MHz_Meteor_Radar. The TDF meteor radar data can be requested from DJ (diego.janches@nasa.gov). The KSS meteor radar data is available on request from YHK (yhkim@cnu.ac.kr).

## Appendix A

## A1 Holographic analysis

The holograms shown in Figure A1 are computed from the semidiurnal tidal phases in the complex domain to account for phase wrapping. As a time dependent phase corresponds to a frequency shift, it is straight forward to estimate the potential Doppler shift of the tidal frequency. Colors towards the red indicate a shift towards lower frequencies, whereas a shift towards the blue indicate higher frequencies in analogy to optical Doppler measurements. The method is presented in more detail in Stober
et al. (2019). It is evident from this figure that the natural variability of the semidiurnal tide already covers the frequency of the predicted lunar tide during a Pekeris resonance (12.42 hours) (Forbes and Zhang, 2012; Zhang and Forbes, 2014). However, these frequent phase shifts appear to be uncorrelated with the lunar orbit and can occur throughout the year at all meteor radar locations without satisfying the vertical conditions of the winds and temperature fields required for the Pekeris resonance.

## A2 Gravity wave residual kinetic energies

The Reynolds decomposition with the ASF can also be used to separate the resolved gravity waves from tides and the daily mean winds. Figure A2 shows the gravity wave residuals after subtracting the daily mean zonal and meridional wind, as well as the diurnal and semidiurnal tides. This gravity wave residual essentially contains the inertia gravity waves with periods up to 16 hours that are not tides. The separation between tide and gravity wave is done by the vertical wavelength.

## A3 Reynolds decomposition comparison of ASF and harmonic fitting

The Reynolds decomposition is a very important element in the momentum flux and wind variance measurements. Therefore, we tested different methods to perform the Reynolds decomposition. Figure A3 shows a comparison of the ASF (upper two panels), a harmonic fit with 24 hours window (central panels) and the harmonic fit with 5 day window (lower two panels). The left column always shows the reconstructed time series using a daily mean winds and the diurnal, semidiurnal and terdiurnal tide. The right column shows the resulting gravity wave residuals. The 2D ASF performs well in recovering a very smooth tidal field and even changes in the tidal phases with altitude. The main difference compared to the 24-hour fit is given in a cleaner

**Figure A1.** Holographic analysis of the semidiurnal tidal variability and lunar elevation angle (thin black line) for each station during the year 2019 for a) TDF, b) KEP, c) King Sejong Station, d) ROT, e) Davis and f) MCM.




removal of a potential inertia gravity wave contamination of the fitted tidal fields due to gravity waves with rather short vertical wavelengths. A 5-day harmonic fit increases the gravity wave residuals significantly, as phase variations of the tide within the long window are no longer sufficiently captured.

*Author contributions.* The conceptional idea of this study was developed by GS, DJ and DF in the frame of the SOUTHTRAC campaign.
VM and KB substantially helped with the data analysis of MLS and the development of the ASF. JM contributed to the data analysis at MCM. All authors reviewed and edited the manuscript. TMG, NM, DM, DJ, JM, SP, WL and YHK provided meteor radar data.

*Competing interests.* The authors declare that there are no competing interests.

*Acknowledgements.* DJ was supported by by the NASA NASA Heliophysics ISFM Program. TDF's operation is supported by NASA SSO, NESC assessment TI-17- 01204, and NSF grant AGS-1647354. For NM and TMG this work was supported by the Natural Environment Re-
search Council [grant numbers: NE/R001391/1, NE/R001235/1]. YHK and WL are financially supported by Korea Polar Research Institute. Operation of the Davis meteor radar was supported under Australian Antarctic Science project 4445. The authors appreciate the invaluable support of the EARG personnel with the operation of TDF. GS is a member of the Oeschger Center for Climate Change Research. GS acknowledges the helpful discussions within the DFG research unit MS-GWaves.

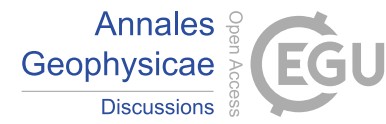

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





**Figure A2.** Kinetic gravity wave energy as determined for the resolved waves from the ASF for each station during the year 2019 for a) TDF, b) KEP, c) KSS, d) ROT, e) DAV and f) MCM.



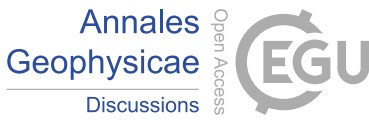

# ASF decomposition (2D)

**a)** RioGrande mean + tides 2019      **b)** RioGrande gravity waves 2019

# 24-hour harmonic fit

**c)** RioGrande mean + tides 2019      **d)** RioGrande gravity waves 2019

# 5-day harmonic fit

**e)** RioGrande mean + tides 2019      **f)** RioGrande gravity waves 2019

**Figure A3.** Comparison of different Reynolds decomposition for 8 days at TDF (RioGrande) to underline the sensitivity on the applied method to derive the gravity wave fluctuations. Panel a) shows the ASF reconstructed daily mean plus diurnal, semidiurnal and terdiurnal tide time series. Panel b) presents the gravity wave residuals. Panels c) and d) show the same, but for a 24-hour harmonic fit and panels e) and f) for a 5 day harmonic fit.