# Peer review of "Seasonal evolution of winds, atmospheric tides and Reynolds stress components in the Southern hemisphere mesosphere/lower thermosphere in 2019"

_Annales Geophysicae, 2020_

## Referee Comment (RC1) · Anonymous Referee #1 · 14 Sep 2020

Reviewer comments

MS No.: angeo-2020-55 Seasonal evolution of winds, atmospheric tides and Reynolds stress components in the Southern hemisphere MLT in 2019 by Gunter Stober et al.

Summary evaluation:

This paper presents an analysis of data collected from measurements of six meteoric radars in the mid- to high-latitude southern hemisphere. The measurements are related to the height range of 75-105 km. The analysis aims to separate the mean winds, tides,

and residual fluctuations, the latter representing gravity waves (GW) and considered as Reynolds stress components. One-year-long time series of each parameter from each of the radar are constructed revealing characteristic seasonal patterns at each location. The momentum fluxes and wind variances are calculated using a recently developed algorithm (described elsewhere) which allows more accurate decomposition of the measured signal into a background flow and the GW fluctuations. Considerable longitudinal and latitudinal asymmetry (or variability) is found in the wintertime zonal and meridional wind patterns. Because a minor stratospheric warming occurred in the southern hemisphere in 2019, the observed asymmetry is believed to be related to the asymmetric structure of the polar vortex at mesospheric heights forming spatially variable filter for vertical GW propagation.

The study introduces new multi-radar observations, utilizes sophisticated data retrieval methods, and contributes to a better understanding of the mesosphere-low thermosphere circulation and vertical coupling in the atmosphere. The paper seems significant and appropriate for publication but substantial revision is needed.

The paper is interesting to read but this referee met some inconsistencies while reading. The following questions and comments should be addressed and clarified in the revised version.

Major comments.

First of all, in some way, the paper is not clearly and concisely written. In particular:

1) Introduction (ll. 16-76). This section seems imbalanced regarding the topic indicated in the title. It is focused on the gravity waves only, while the title (and the study as a whole) includes winds and tides. Briefly: the first part of the introduction discusses the role of GWs in the atmospheric dynamics (ll. 16-46), the second part (ll. 47-63) discusses the advantage of the meteor radar in the GW detection along with the techniques applied to retrieve the wind fluctuations, the rest of the section (ll. 64-76) presents the list of the observational sites and introduces the structure of the paper.

The mean winds and tides in the southern hemisphere should be introductorily discussed in more detail in order to put the present study into context in this regard.

It is worth mentioning (in the introduction or anywhere throughout the paper) what approximate fraction of the energy is carried by the GWs with periods less than those detected by the meteor radar (<1 h)?

2) Since comparisons are always made between stations, but not between components, it seems reasonable to rearrange Figs 2-6, 8, 9 so that zonal and meridional (and vertical) parameters from different stations are presented in separate plots. In this case, it is easier to visually identify spatial similarities and differences. Please consider rearranging.

3) Conclusion. This section (57 lines in total) is too broad to be a proper conclusion. A great part of the material included is more appropriate for the discussion section. There are even several references. The section should be shorter and more focused. In its present form, it is difficult to take the main new results of the study. Here (or in the introduction), the main goal of the paper should be explicitly formulated.

Lines 509-510. In the present paper a new retrieval algorithm is not presented but rather utilized, isn't it?

4) Fig. 5. It is curious why the spring semidiurnal tidal amplitude at KEP is much larger than at all other stations. Is there any explanation? Other parameters do not show any considerable difference for KEP.

5) Lines 453-458. The methodological shortcoming of this statement is that, based on one particular year during which a SSW occurred and without comparison with non-SSW years, we are not confident that the observed variability in the MLT winds is related to a wobbling and asymmetric polar vortex. The statement should be taken with caution. Maybe, the typical inter-station variability is of the same order.

Minor comments

1) Title. Since the broad geophysical community of AnnGeo readers may not be very familiar with the acronym MLT, it seems better to use the full name: mesosphere-low thermosphere. 2) For convenience, please indicate the radar codes in Fig. 1 (e.g., in the right panel) and include the full name of the sites to Table 1. 3) Line 186. Correct reference?

---

## Referee Comment (RC2) · Anonymous Referee #1 · 14 Sep 2020

Please discard minor comment # 1 (title), I'm sorry for misreading.

---

## Referee Comment (RC3) · Anonymous Referee #2 · 21 Sep 2020

Reviewer comments:

This paper describes an approach to obtain wind variances and momentum fluxes. In which an adaptive spectral filter has been used to perform the Reynolds decomposition into a background flow and the GW fluctuations. The authors have used winds obtained during 2019 by 6 meteor radars from middle to polar latitudes in southern hemisphere. To reduce the meteor altitude uncertainty, a full earth geometry was implemented, which maximize the observed number of meteors in the analysis. The topic covered in the manuscript is important as it contributes to improving the momentum flux and

variance estimates in MLT region winds. The arguments used to interpret the results are not clear and sometimes not convincing. The scientific contribution is appropriate for this journal. However, there are some issues that need to be addressed.

Comments:

In the "Introduction", some sources of secondary gravity waves have been emphasized, so it was expected that the authors would also explore this knowledge in the results as well as in the discussion. In this sense, there is a lack of enough discussion about this topic. Some parts of the description of the Reynolds stress results are confusing. For example:

Line 399 - "In Particular, at KSS a variable zonal momentum flux is measured that seems to be in better agreement with TDF and ROT results". For me, it is hard to see a better agreement among KSS and TDF/ROT results, from Figure 8.

Lines 404-405 - For the "results from KSS and KEP show a good agreement of the vertical structure ...", from the Figure 8, I can see that a good agreement occur above 90 km.

Lines 405-407 - where appear "Results from ROT and DAV still reflect some features of the seasonal meridional momentum flux behaviour," again, from the Figure 8 it is possible to see that KEP "still reflect some features of ..." - instead of DAV.

Discussion should be made more rigorous. The basis for these statements need to expand further, considering the stratospheric and MLT winds (Figures 10 and 2) to explain the momentum flux components and variations observed (Figures 8 and 9). What configurations are expected for momentum flux in face of the observed stratospheric and MLT winds?

Technical revision

Line 23: Change (Fritts and Nastrom (1992) by "Fritts and Nastrom (1992)" Line 26: "ch" and "Uh" don't seem necessary. Lines 398-399: change "In Particular, at KSS a" by
"In particular, at KSS a" Lines 471 and 473: it is unnecessary to use the acronym USLM

Line 496: change "structure Becker and Vadas (2018) with" by "structure (Becker and Vadas, 2018) with"

---

## Author Comment (AC1) · 5 Oct 2020

General comment:

Anonymous Referee #1

Seasonal evolution of winds, atmospheric tides and Reynolds stress components in the Southern hemisphere MLT in 2019 by Gunter Stober et al. Summary evaluation: This paper presents an analysis of data collected from measurements of six meteoric radars in the mid- to high-latitude southern hemisphere. The measurements are related to the height range of 75-105 km. The analysis aims to separate the mean winds, tides, and residual fluctuations, the latter representing gravity waves (GW) and considers Reynolds stress components. One-year-long time series of each parameter from each of the radar are constructed revealing characteristic seasonal patterns at each location. The momentum fluxes and wind variances are calculated using a recently developed algorithm (described elsewhere) which allows more accurate decomposition of the measured signal into a background flow and the GW fluctuations. Considerable longitudinal and latitudinal asymmetry (or variability) is found in the wintertime zonal and meridional wind patterns. Because a minor stratospheric warming occurred in the southern hemisphere in 2019, the observed asymmetry is believed to be related to the asymmetric structure of the polar vortex at mesospheric heights forming spatially variable filter for vertical GW propagation. The study introduces new multi-radar observations, utilizes sophisticated data retrieval methods, and contributes to a better understanding of the mesosphere-low thermo-sphere circulation and vertical coupling in the atmosphere. The paper seems significant and appropriate for publication but substantial revision is needed. The paper is interesting to read but this referee met some inconsistencies while reading. The following questions and comments should be addressed and clarified in the revised version.

General Reply:

We thank the reviewer for his suggestions which greatly improved the quality of the manuscript. Following these suggestions, we have updated all the figures and the corresponding paragraphs to keep the logical order. Further, we expanded the introduction and shortened the conclusion. The changes are indicated with latexdiff.

Comment:

3) Conclusion. This section (57 lines in total) is too broad to be a proper conclusion. A great part of the material included is more appropriate for the discussion section. There are even several references. The section should be shorter and more focused.

In its present form, it is difficult to take the main new results of the study. Here (or in the introduction), the main goal of the paper should be explicitly formulated. Lines 509-510. In the present paper a new retrieval algorithm is not presented but rather utilized, isn't it?

Reply:

We rewrote the introduction as well- as the conclusion to provide a more focused manuscript. The detailed changes are indicated by colors using latexdiff in the revised manuscript.

Comment:

4) Fig. 5. It is curious why the spring semidiurnal tidal amplitude at KEP is much larger than at all other stations. Is there any explanation? Other parameters do not show any considerable difference for KEP.

Reply:

A detailed investigation of why the semidiurnal tide during spring is amplified at KEP, but not as strong at the other stations is beyond the scope of this manuscript and a thorough investigation will require stratospheric data as well. Currently, merged data sets of model outputs and observations are prepared, but not yet available. However, it is the meridional component that is much stronger at KEP and seems to be related to the mean meridional wind, which shows only at KEP a strong southward wind during this time. The other stations at TDF, KSS and ROT show a reversal of the meridional component towards northward winds during this period. However, at this time we cannot confirm that this is the explanation for the tidal amplitude differences or whether there are additional effects from the stratosphere also playing a critical role. Further, global data sets are required to investigate migrating and non-migrating tidal components and how their forcing is affected by mean winds in the stratosphere, which might also help to understand the longitudinal differences. We added a reference to Murphy

et al., 2006, who investigated the non-migrating tides at the SH.

Comment:

5) Lines 453-458. The methodological shortcoming of this statement is that, based on one particular year during which an SSW occurred and without comparison with non-SSW years, we are not confident that the observed variability in the MLT winds is related to a wobbling and asymmetric polar vortex. The statement should be taken with caution. Maybe, the typical inter-station variability is of the same order.

Reply:

The statement refers to the climatological mean at TDF, which was compiled from the years 2008 to 2018. However, since the manuscript focuses on 2019, we added a statement underlining that the wobbling nature was found for 2019 and we refer to the climatology at TDF between 2008-2018. This is now explicitly mentioned in the revised manuscript. We also looked at the polar vortex asymmetry in the MLS climatology and found again a consistent pattern. This suggests that 2019 concerning the seasonal behavior was not an exceptional. Only due to the occurrence of the minor SSW it become rather unique.

Minor Comments:

Comment:

1)Title. Since the broad geophysical community of AnnGeo readers may not be very familiar with the acronym MLT, it seems better to use the full name: mesosphere-low thermosphere.

Reply:

This comment was withdrawn.

Comment:

[Figure]

2) For convenience, please indicate the radar codes in Fig. 1 (e.g., in the right panel) and include the full name of the sites to Table 1.

Reply:

Done.

Comment:

3) Line 186. Correct reference?

Reply:

A first version of the ASF(1D) is already described in Stober et al., 2017, although we did not introduce the acronym ASF in this paper and only present a brief description of the algorithm. The paper already mentions adaptive spectral filter. This is described at the end of section 2.1.

---

## Author Comment (AC2) · 5 Oct 2020

General comment:

Anonymous Referee #2

This paper describes an approach to obtain wind variances and momentum fluxes. In which an adaptive spectral filter has been used to perform the Reynolds decomposition into a background flow and the GW fluctuations. The authors have used winds obtained during 2019 by 6 meteor radars from middle to polar latitudes in southern hemisphere.

To reduce the meteor altitude uncertainty, a full earth geometry was implemented, which maximize the observed number of meteors in the analysis. The topic covered in the manuscript is important as it contributes to improving the momentum flux and variance estimates in MLT region winds. The arguments used to interpret the results are not clear and sometimes not convincing. The scientific contribution is appropriate for this journal. However, there are some issues that need to be addressed.

General Reply:

We thank the reviewer for his comments and suggestions. During the preparation of the manuscript, we emphasized on the technical details, which somehow led to a too short discussion of some scientific aspects. In response to these comments, we expanded the scientific discussion in the suggested context. All changes of the manuscript will be indicated by latexdiff.

Comments:

In the "Introduction", some sources of secondary gravity waves have been emphasized, so it was expected that the authors would also explore this knowledge in the results as well as in the discussion. In this sense, there is a lack of enough discussion about this topic. Some parts of the description of the Reynolds stress results are confusing.

Reply:

We appreciate this suggestion and expand the discussion for the Antarctic Peninsula and McMurdo as these non-primary waves show remarkably agreement with the observed momentum fluxes and variances. The 2D ASF seems to be very suitable to provide a much better filtering for non-primary waves compared to temporal-only filters.

Comment:

For example: Line 399 - "In Particular, at KSS a variable zonal momentum flux is measured that seems to be in better agreement with TDF and ROT results". For me, it

is hard to see a better agreement among KSS and TDF/ROT results, from Figure 8.

Reply:

This point is also related to a comment from reviewer #1 and following both reviewers' suggestion, we sorted the figures with respect to each meteorological parameter for each station, which enables a more straightforward inter-station comparison. This was accompanied by changing the order of some related paragraphs to keep a logical order between the Figures and the text.

Comment:

Lines 404-405 - For the "results from KSS and KEP show a good agreement of the vertical structure ...", from the Figure 8, I can see that a good agreement occur above 90 km.

Reply:

This sentence was rephrased.

Comment:

Lines 405-407 - where appear "Results from ROT and DAV still reflect some features of the seasonal meridional momentum flux behaviour," again, from the Figure 8 it is possible to see that KEP "still reflect some features of ..." - instead of DAV. Discussion should be made more rigorous. The basis for these statements need to expand further, considering the stratospheric and MLT winds (Figures 10 and 2) to explain the momentum flux components and variations observed (Figures 8 and 9). What configurations are expected for momentum flux in face of the observed stratospheric and MLT winds

Reply:

We added a paragraph to discuss in more detail the observed momentum fluxes and wind variances with respect to the observed mean winds in 2019 for the different stations.

Technical revision

Comment:

Line 23: Change (Fritts and Nastrom (1992) by "Fritts and Nastrom (1992)"

Reply:

Done.

Comment:

Line 26: "ch" and "Uh" don't seem necessary.

Reply:

Done.

Comment:

Lines 398-399: change "In Particular, at KSS a" by "In particular, at KSS a"

Reply:

Done.

Comment:

Lines 471 and 473: it is unnecessary to use the acronym USLM

Reply:

Done.

Comment:

Line 496: change "structure Becker and Vadas (2018) with" by "structure (Becker and Vadas, 2018) with"

Reply:

Done.

---

## Author Response (AR2)

Technical corrections:

Seasonal evolution of winds, atmospheric tides and Reynolds stress components in the Southern hemisphere mesosphere/lower thermosphere in 2019

Dear Editor,

the finally uploaded manuscript is including the following technical corrections.

- The typo is corrected.
- Figures 3-6 are updated according to the reviewer suggestion, Figure is kept as the redundancy in the title and axis label is minor.

Best regards,
Gunter